# Evolution of the Tumor Microenvironment toward Immune-Suppressive Seclusion during Brain Metastasis of Breast Cancer: Implications for Targeted Therapy

**DOI:** 10.3390/cancers13194895

**Published:** 2021-09-29

**Authors:** Myung-Giun Noh, Sung Sun Kim, Yeong Jin Kim, Tae-Young Jung, Shin Jung, Joon-Haeng Rhee, Jae-Hyuk Lee, Ji-Shin Lee, Jae-Ho Cho, Kyung-Sub Moon, Hansoo Park, Kyung-Hwa Lee

**Affiliations:** 1Department of Biomedical Science and Engineering, Gwangju Institute of Science and Technology (GIST), Gwangju 61005, Korea; gabriel0421@gist.ac.kr or; 2Department of Pathology, Chonnam National University Hwasun Hospital and Medical School, Chonnam National University Research Institute of Medical Science, BioMedical Sciences Graduate Program (BMSGP), Hwasun 58128, Korea; succeedsoon@hanmail.net (S.S.K.); jhlee@jnu.ac.kr (J.-H.L.); jshinlee@jnu.ac.kr (J.-S.L.); 3Department of Neurosurgery, Chonnam National University Hwasun Hospital and Medical School, Chonnam National University Research Institute of Medical Science, Hwasun 58128, Korea; mbdosa88@naver.com (Y.J.K.); jung-ty@chonnam.ac.kr (T.-Y.J.); sjung@chonnam.ac.kr (S.J.); moonks@chonnam.ac.kr (K.-S.M.); 4Medical Research Center for Combinatorial Tumor Immunotherapy, Department of Microbiology and Immunology, Chonnam National University Medical School, Hwasun 58128, Korea; jhrhee@jnu.ac.kr (J.-H.R.); jh_cho@jnu.ac.kr (J.-H.C.); 5Immunotherapy Innovation Center, Chonnam National University Hwasun Hospital and Medical School, Hwasun 58128, Korea

**Keywords:** breast neoplasms, tumor microenvironment, neoplasm metastasis, gene expression profiling, immunohistochemistry

## Abstract

**Simple Summary:**

Brain metastases (BM) of breast cancer (BC) are new targets of immunotherapy, but their characteristics are unclear. Therefore, we analyzed the differential expression profile of the tumor microenvironment (TME) in primary breast cancer brain metastasis (BCBM). In the TME of BCBM, immune-related pathways were downregulated and tumor intrinsic factors were upregulated. Moreover, CD8+ T cells and M1 macrophages with cytotoxic effects were decreased, but M2 cells were increased, in BM. Most tumor-suppressive immune functions ceased after BM with a molecular subtype shift. These results suggest the need for targeted therapy and immunotherapy strategies for BCBM.

**Abstract:**

Breast cancer (BC) is the second most common solid malignant tumor that metastasizes to the brain. Despite emerging therapies such as immunotherapy, whether the tumor microenvironment (TME) in breast cancer brain metastasis (BCBM) has potential as a target of new treatments is unclear. Expression profiling of 770 genes in 12 pairs of primary BC and matched brain metastasis (BM) samples was performed using the NanoString nCounter PanCancer IO360^TM^ Panel. Immune cell profiles were validated by immunohistochemistry (IHC) in samples from 50 patients with BCBM. Pathway analysis revealed that immune-related pathways were downregulated. Immune cell profiling showed that CD8+ T cells and M1 macrophages were significantly decreased, and M2 macrophages were significantly increased, in BM compared to primary BC samples (*p* = 0.001, *p* = 0.021 and *p* = 0.007, respectively). *CCL19* and *CCL21*, the top differentially expressed genes, were decreased significantly in BM compared to primary BC (*p* < 0.001, both). IHC showed that the CD8+ count was significantly lower (*p* = 0.027), and the CD163+ and CD206+ counts were higher, in BM than primary BC (*p* < 0.001, both). A low CD8+ T cell count, low CD86+ M1 macrophage count, and high M2/M1 macrophage ratio were related to unfavorable clinical outcomes. BC exhibits an immunosuppressive characteristic after metastasis to the brain. These findings will facilitate establishment of a treatment strategy for BCBM based on the TME of metastatic cancer.

## 1. Introduction

Breast cancer (BC) is the second most common solid cancer that metastasizes frequently to the central nervous system (CNS), following lung cancer [1]. Brain metastasis (BM) occurs in up to 30% of patients with BC and is a major clinical issue, with a 15-month median overall survival (OS) [2,3]. Thus, BM from BC (breast cancer and brain metastasis, BCBM) lowers the quality of life of BC patients. Understanding the biological traits of BCBM is important for assessing prognosis and identifying new therapeutic targets.

In general, patients with BM can be treated by whole-brain radiotherapy (WBRT), tumor resection, or stereotactic radiosurgery, such as gamma knife radiosurgery, considering the multiplicity and resectability of the brain metastatic lesion [4,5]. In recent decades, with the advent of innovative targeted therapies and immunotherapies, the archetype of BM treatment has begun to move toward systemic from localized treatment modalities [5]. Immune checkpoint inhibitors (ICIs), such as anti-programmed cell death protein 1 (PD-1)/programmed death-ligand 1 (PD-L1) antibodies, have revolutionized cancer therapy [6]. Treatment with ICIs is associated with improved response rates in many malignancies, including melanoma, non-small cell lung cancer (NSCLC), head-and-neck squamous cell cancer, advanced gastric cancer, and renal cell carcinoma [7,8]. Unfortunately, BC is less amenable to treatment with ICIs due to its inherently low immunogenicity [9]. Combinatorial therapy with the PD-L1 inhibitor atezolizumab and nanoparticle albumin-bound (nab) paclitaxel is now the standard first-line therapy in patients with metastatic triple-negative breast cancer (TNBC) and a PD-L1–positive immune population [10]. However, this is only approved for a small number of patients, and further research is needed to expand the indications for this treatment modality.

The tumor microenvironment (TME) plays a pivotal role in tumorigenesis and progression via close interactions with tumor cells [11]. The TME is composed of cellular and non-cellular compartments. Cancer cells, immune cells, blood vessels, and lymphatics, and fibroblasts comprise the cellular portion of the TME. The non-cellular portion encompasses cytokines, chemokines, mediators, and growth factors [12,13]. The TME is necessary for invasion, metastasis, and settling in a distant location [14,15]. TME also mediates the process by which tumors involve the CNS [16]. There are innate and adaptive immune cell types in brain tumors, similar to other tumor types. Antitumor T-cell responses were inhibited by highly immunosuppressive brain TMEs, even in the context of ICIs [17]. The efficacy of ICIs in a melanoma mouse model depended on the presence of extracranial tumors and increased CD8+ T-cell trafficking into BM [18]. The necessity of CD8+ T-cell priming and trafficking to CNS lesions, to mount antitumor immune responses in synergy with ICIs, was illustrated by vascular endothelial growth factor-C (VEGF-C)-induced modulation of the meningeal lymphatic system [19]. Therefore, immune modulation in BM is an obstacle to overcoming resistance to ICIs [20,21]. The TME of BCBM is poorly understood. An in-depth understanding of the biological and immunological characteristics of the TME in BCBM could lead to the discovery of new targets for immunotherapy and improvement of therapeutic outcomes. 

We evaluated the TME of BCBM by comparative gene expression profiling of primary BC. Using the NanoString nCounter PanCancer IO 360™ Panel, we analyzed the differentially expressed genes (DEGs), immune profiles, and functional signature of immune regulation. We also performed IHC to validate these immune aspects. Collectively, we investigated the TME in primary BC and BCBM and its mechanisms, and comparatively analyzed prognosis according to differential immune factors.

## 2. Materials and Methods

### 2.1. Sample Collection

Formalin-fixed paraffin-embedded (FFPE) samples were collected from 50 patients who underwent removal of BCBM at Chonnam National University Hwasun Hospital between 2004 and 2020. Of the 50 patients, 5 were operated on twice for initial and recurrent BM. Matched primary BC samples were available for 24 patients, either in core needle biopsies or surgical samples. Hematoxylin and eosin-stained tissue slides were reviewed, and representative tissue FFPE blocks were selected by three pathologists (M.-G.N., S.S.K., and K.-H.L.). Of them, 12 matched pairs were selected, with a preference for those with matched primary cancer surgical samples, and for more recent cases. Clinical data were retrospectively collected from the patients’ medical records. For cases without matched primary BC tissues, information on estrogen receptor (ER), progesterone receptor (PR), and human epidermal growth factor receptor 2 (HER2) status at the time of pathologic diagnosis was collected from the medical records. Immunohistochemistry (IHC) for ER, PR, and HER2 was performed for cases with available tissue samples. The size of the tumor was measured as the greatest diameter during the gross examination. This study was approved by the Institutional Review Board of Chonnam National University Hwasun Hospital (CNUHH–2019–218).

### 2.2. Gene Expression Assay Using the NanoString nCounter PanCancer IO 360™ Panel

Gene expression was measured using the NanoString nCounter PanCancer IO 360™ Panel (NanoString Technologies, Seattle, WA, USA). The PanCancer IO 360™ Panel consists of 770 genes, including 20 housekeeping genes. Twelve pairs of matched primary BC and BCBM samples were subjected to gene expression assay. For better RNA quality, the most recently acquired samples were selected from each group based on the testing time point. Tissue samples were placed on glass slides as 10-μm thick FFPE sections and subjected to RNA extraction and analysis by PhileKorea Technologies (Seoul, Korea). The analysis of gene expression was conducted on the nCounter^®^ PanCancer IO 360™ Panel and NanoString (NanoString Technologies) platform. The raw transcriptome data were subjected to housekeeping-gene normalization using the geNorm algorithm in nCounter Advanced Analysis ver. 2.0.115 (NanoString Technologies) [22]. Normalized data were log2-transformed for analysis. A quality check of raw data was conducted using nSolver Analysis Software ver. 4.0 and NanoStringQCpro ver. 1.14.0 (NanoString Technologies). 

### 2.3. Differential Gene Expression Analysis

A DEG analysis was carried out using nSolver Analysis Software ver. 4.0.7., to calculate the fold-differences and *p*-values between the primary BC and BCBM groups. R-project ver. 4.0.2 (R Development Core Team, Vienna, Austria) for Mac OS was used for visualization of DEG analysis. For hierarchical clustering of the cases, a correlation analysis was performed using the *cor()* function in R. To plot the principal component analysis (PCA) data, we used the *prcomp()* function in R. Using the EnhancedVolcano package in R, we selected genes with a log_2_ value indicating a fold difference greater than 2 and *p*-value < 0.05. Graphs were generated using the ggplot package in R. 

### 2.4. Pathway Analysis

A gene set enrichment analysis (GSEA) was conducted using the Molecular Signatures Database (MSigDB) ver. 7.4 and GSEA software ver. 4.1.0 (Broad Institute, Cambridge, MA, USA) [23,24]. Results were considered significant at a *p*-value < 0.05 and false discovery rate (FDR) *q*-value < 0.2. For single-sample GSEA (ssGSEA) analysis, gene set variation analysis (GSVA) was performed in R. For pathway analysis of Cancer-Immunity Cycle annotations and functional pathways, the reference data file annotated with the PanCancer IO 360™ Panel (770 genes) was downloaded from the NanoString Technologies website (https://www.nanostring.com/wp-content/uploads/2021/01/LBL-10498-02_IO_360_Gene_List.xlsx, accessed on 9 May 2021). These reference data were processed to derive gene sets in the gene matrix file format (.gmt) for GSEA and GSVA. 

### 2.5. Tumor Immune Cell Deconvolution

The immune cell composition of tumor samples was characterized by nCounter^®^ PanCancer IO360 gene expression panel analysis, the microenvironment cell populations-counter (MCP-counter) method, and quantification of the tumor immune contexture from a human RNA-seq data (quanTIseq) deconvolution algorithm [25,26,27]. The nCounter^®^ method based on nSolver 4.0 was used to evaluate the abundance of 14 immune-cell populations according to the expression levels of cell type-specific marker genes (natural killer [NK] cells, macrophages, neutrophils, CD56dim NK cells, B cells, type 1 helper T (TH1) cells, mast cells, dendritic cells, regulatory T cells (Tregs), CD8+ T cells, CD45+ cells, exhausted CD8+ T cells, cytotoxic cells, and T cells) [28]. MCP-counter produced the absolute abundance score for eight major immune-cell types (CD3+ T cells, CD8+ T cells, cytotoxic lymphocytes, NK cells, B lymphocytes, monocytic lineage cells, myeloid dendritic cells, and neutrophils). The quanTIseq algorithm computed the relative abundances of 10 immune-cell types (B cells, M1 macrophages, M2 macrophages, monocytes, neutrophils, NK cells, CD4+ T cells, CD8+ T cells, Tregs, and dendritic cells). 

### 2.6. Immunohistochemistry

Tissue sections were sectioned at 3 μm thickness and subjected to IHC using a Bond-Max Autostainer (Leica Microsystems, Buffalo Grove, IL, USA), as described previously [29]. The following antibodies were used: CD8 (1:300 dilution; catalogue no. IS623; DAKO, Santa Clara, CA, USA), CD86 (1:150 dilution; E2G8P; Cell Signaling, Danvers, MA, USA), CD163 (1:300 dilution; NCL–CD163; Novocastra, Newcastle-upon-Tyne, UK), CD206 (1:200 dilution; E2L9N; Cell Signaling), and CCL19 (1:20,000 dilution; LS–C798145–100; LSbio, Seattle, WA, USA). IHC was performed using QuPath, which is optimized for quantitative analysis of digital pathology images [30]. For the intensity threshold parameters of QuPath, CD8 was set to “Nucleus: DAB OD mean”; CD86 was set to “Cytoplasm: DAB OD max”; and CD163 and CD206 were set to “Cytoplasm: DAB OD means.” The positive threshold was adjusted from 0.16 to 0.28 based on the differences in background staining intensities among the slides. T cells and macrophages were evaluated within the borders of the invasive tumors; areas outside the tumor border, or around DCIS and normal lobules, were excluded. CD8 was evaluated in lymphocytes only, and CD86, CD163, and CD206 were evaluated only in cells of monocytoid/macrophage-like morphology. After reviewing stained slides from each case, a field of maximum intensity of expression was selected. CD8, CD86, CD163, and CD206 expression was determined by counting the stained cells. In each case, the two visual fields with the highest density of positive cells were selected at a magnification of 200× (1.08 mm^2^). The numbers of stained cells in the two visual fields were averaged. 

IHC of ER (clone SP1, catalog no. 790–4324; Roche Diagnostics, Indianapolis, IN, USA), PR (clone 1E2, catalog no. 790–2223, Roche Diagnostics), and HER2 (clone 4B5, catalog no. 790–4493, Roche Diagnostics) was carried out using Ventana Benchmark Ultra Automated Stainer (Ventana Medical Systems, Tucson, AZ, USA) following the standard protocol. Hormone receptor staining was interpreted as positive if ≥1% of tumor cells showed nuclear staining of any intensity, and as negative if <1% or 0% of tumor cells had nuclear staining [31]. Regarding HER2 interpretation, only cases with a score of 3+ for circumferential membranous staining in >10% of tumor cells were considered positive. No additional in situ hybridization analysis was performed. 

### 2.7. Statistical Analysis

To compare the two matched groups, the Wilcoxon signed-rank test was performed using the *wilcox.test()* function in R. For linear correlations, Pearson correlation analysis was performed using the *ggscatter()* function in R. Survival analysis was performed in R, with optimal cut-point analysis performed using the *surv_cutpoint()* function in the R-package Survminer (ver. 0.4.8) [32]. All statistical analyses were performed using R-project 4.0.2 for Mac OS. *p*-values < 0.05 were considered significant. 

## 3. Results

### 3.1. Patients’ Characteristics

Of the 50 patients with BCBM, clinicopathological data were available for 44 cases of primary BC and 50 cases of BCBM. The median ages of the patients were 47 years at diagnosis of primary BC and 54 years at diagnosis of BCBM. The median diagnostic interval from primary BC to BM was 3 years. The median tumor sizes in primary BC and in BCBM were 2.4 and 4.15 cm, respectively. In primary BC, 20 (45.5%) of patients were positive for ER, 16 (36.4%) were positive for PR, and 23 (52.3%) were positive for HER2. In BCBM, 17 (34.0%) cases were positive for ER, 9 (18.0%) were positive for PR, and 25 (50.0%) were positive for HER2. Alteration of ER, PR, and HER2 status and molecular subtype switching during BM in patient-matched cases were visualized using river plots (Appendix A). The clinicopathological profiles of the primary BC and BCBM cases are listed in Table 1.

### 3.2. Differential Gene Expression in Primary BC and BM

Twelve matched pairs of primary BC and BCBM were analyzed by NanoString gene expression assay. To assess the transcriptional similarity between cases, we performed unsupervised hierarchical clustering of normalized gene expression values (Figure 1a). The cases of primary BC and BCBM were divided into different groups. A principal components analysis showed that individual cases were classified into two groups according to location (Figure 1b). Of the 750 genes in the nCounter^®^ PanCancer IO360 gene expression panel, *CCL19* and *CCL21* showed the highest differential expression between primary BC and BCBM (5.82- and 4.72-fold change, respectively; adjusted *p* < 0.001, both) (Figure 1c). A heatmap showed the differential expression profile of primary BC and BCBM. Unsupervised hierarchical clustering of the differential expression levels of genes revealed that primary BC and BCBM were classified into separate clusters (Figure 1d). In the individual-patient BC-BM pair analysis, the expression levels of *CCL19*, *CCL21*, and *ESR1* in BCBM were lower than those in primary BC (*p* < 0.001, *p* < 0.001, and *p* = 0.003, respectively) (Figure 1e). By contrast, the expression of *ERBB2* was significantly higher in BCBM (*p* = 0.012).

### 3.3. Comparative Pathway Analysis

To compare the TME immune response between primary BC and BCBM, we performed pathway analysis of Cancer-Immunity Cycle annotations from the nCounter^®^ PanCancer IO 360™ Panel using ssGSEA. Unsupervised hierarchical clustering of each case showed that primary BC and BCBM formed separate clusters (Figure 2a). GSEA of cancer-immunity cycle annotations revealed that primary BC was significantly enriched in genes related to immune cell localization to tumors (normalized enrichment score (NES) = 1.70, nominal *p*-value (NOM *p)* < 0.001, false discovery rate *q*-value (FDR *q*) < 0.001). Recognition of cancer cells by T cells (NES = 1.41, NOM *p* = 0.005, FDR *q* = 0.026), myeloid cell activity (NES = 1.41, NOM *p* < 0.001, FDR *q* = 0.018), stromal factors (NES = 1.35, NOM *p* = 0.007, FDR *q* = 0.030), T-cell priming and activation (NES = 1.38, NOM *p* = 0.001, FDR *q* = 0.021), cancer antigen presentation (NES = 1.34, NOM *p* = 0.008, FDR *q* = 0.026), and NK cell activity (NES = 1.29, NOM *p* = 0.111, FDR *q* = 0.048) (Figure 2b). Genes related to the release of cancer cell antigens were upregulated in BCBM compared to primary BC (NES = 2.18, NOM *p* < 0.001, FDR *q* < 0.001). The individual-patient BC-BM pair analysis showed that the scores for cancer antigen presentation, immune cell localization to tumors, stromal factors, NK cell activity, T-cell priming and activation, recognition of cancer cells by T cells, myeloid cell activity, and killing of cancer cells were significantly lower in BCBM than primary BC (*p* < 0.001, *p* < 0.001, *p* < 0.001, *p* < 0.001, *p* = 0.001, *p* = 0.001, *p* = 0.001, and *p* = 0.005, respectively) (Figure 2c). The scores for release of cancer cell antigen, cell cycle and proliferation, immunometabolism, and tumor intrinsic factors were significantly higher in BCBM than primary BC (*p* < 0.001, *p* < 0.001, *p* = 0.001 and *p* = 0.003, respectively) (Figure 2d). We next performed a pathway analysis of functional pathways included in the cancer-immunity cycle annotations from the nCounter^®^ PanCancer IO 360™ Panel using ssGSEA. Unsupervised hierarchical clustering showed that immune-related pathways—antigen presentation, cytokine and chemokine signaling, lymphoid compartment, JAK–STAT signaling, co-stimulatory signaling, immune cell adhesion and migration, myeloid compartment, cytotoxicity, and interferon signaling—clustered together and were more active in primary BC than BCBM (Appendix A). Unsupervised hierarchical clustering showed that tumor stroma-related pathways—angiogenesis, matrix remodeling and metastasis, and tumor intrinsic factors including hypoxia, apoptosis, autophagy, epigenetic regulation, metabolic stress, cell proliferation. and DNA damage repair—clustered together and were upregulated in BCBM. GSEA of functional pathways in the cancer-immunity cycle annotations between primary BC and BCBM revealed that primary BC was significantly enriched in genes related to immune cell adhesion and migration, antigen presentation, lymphoid compartment and co-stimulatory signaling, whereas genes related to epigenetic regulation, cell proliferation, metabolic stress, and DNA damage repair were enriched in BCBM compared to primary BC (Appendix A). The individual-patient BC-BM pair analysis showed that the scores for immune cell adhesion and migration, co-stimulatory signaling, cytokine and chemokine signaling, matrix remodeling and metastases, myeloid compartment, lymphoid compartment, antigen presentation, and interferon signaling were significantly higher in primary BC (*p* < 0.05, for all) (Appendix A). By contrast, cell proliferation, epigenetic regulation, hypoxia, metabolic stress, DNA damage repair, apoptosis and autophagy were significantly enhanced in BCBM (*p* < 0.05, for all) (Appendix A).

### 3.4. Biological Signature Analysis

To evaluate the biological differences of TME crucial to the tumor-immune interaction, we performed a pathway analysis of the biological signature of the nCounter^®^ PanCancer IO 360™ Panel. As with the cancer-immune cycle, primary BC and BCBM formed separate clusters based on unsupervised hierarchical clustering (Figure 2e). The heatmap showed that antigen-presentation-related pathways such as PD-1, PD-L1, *CTLA4*, antigen-presenting machinery, and B7-H3 were downregulated in BCBM compared to primary BC. The individual-patient BC-BM pair analysis showed that most immune-related scores of biologic signatures—antigen-presenting machinery (*p* = 0.001), lymphoid (*p* = 0.001), MHC2 (*p* = 0.001), PD–L2 (*p* = 0.001), stroma (*p* = 0.001), tumor inflammation signature (*p* = 0.002), *CTLA4* (*p* = 0.002), cytotoxicity (*p* = 0.002), myeloid (*p* = 0.002), *TIGIT* (*p* = 0.002), B7–H3 (*p* = 0.003), inflammatory chemokines (*p* = 0.003), *IL10* (*p* = 0.007), myeloid inflammation (*p* = 0.007), PD-1 (*p* = 0.007), interferon (IFN)-gamma (*p* = 0.007), immunoproteasome (*p* = 0.014), *IDO1* (*p* = 0.019), PD-L1 (*p* = 0.017), and IFN downstream (*p* = 0.032)—were significantly decreased in BCBM (Figure 2f). The signature scores of proliferation (*p* = 0.01) and glycolytic activity (*p* = 0.042) were significantly higher in BCBM (Figure 2g). 

### 3.5. Immune Cell Profile Analysis

Next, immune cell composition was evaluated using the nCounter^®^ PanCancer IO360 immune profile panel. Unsupervised hierarchical clustering showed that primary BC and BCBM formed separate clusters (Figure 3a). Immune cell expression tended to be higher in primary BC than BCBM. Individual-patient BC-BM pair analysis showed that the scores of most immune-cell subsets—CD8+ T cells (*p* = 0.001), cytotoxic cells (*p* = 0.001), exhausted CD8+ T cells (*p* = 0.001), T cells (*p* = 0.001), B cells (*p* = 0.001), mast cells (*p* = 0.001), CD45+ cells (*p* = 0.002), dendritic cells (*p* = 0.003), Tregs (*p* = 0.005), TH1 cells (*p* = 0.01), neutrophils (*p* = 0.014), macrophages (*p* = 0.019), and CD56dim NK cells (*p* = 0.042)—were significantly decreased in BCBM compared to primary BC (Figure 3b). Individual-patient BC-BM pair analysis using the MCP counter platform showed that the scores of most immune-cell subsets—T cells (*p* < 0.001), CD8+ T cells (*p* < 0.001), cytotoxicity score (*p* < 0.001), B cells (*p* < 0.001), the ratio of macrophages to monocytes (*p* = 0.001), monocytes (*p* = 0.001) and endothelial cells (*p* = 0.001)—were significantly lower in BCBM than primary BC (Figure 3c). There was no significant difference in NK cells between the nCounter^®^ PanCancer IO360 immune profile and MCP counter (*p* = 0.102 and *p* = 0.519, respectively) analyses. To assess macrophage subtypes, we performed a quanTIseq analysis. Individual-patient BC-BM pair analysis showed that the M2-subtype score was significantly higher in BCBM, while the M1-subtype score was significantly lower in BCBM (*p* = 0.021 and *p* = 0.007, respectively) (Figure 3d). We performed a Pearson linear correlation analysis between *CCL19* expression and the nCounter^®^ PanCancer IO360 immune profile. *CCL19* expression showed a significant association with all 14 immune cell profiles, including CD8+ T cells (*R* = 0.72, *p* < 0.001), cytotoxic cells (*R* = 0.71, *p* < 0.001), macrophages (*R* = 0.72, *p* < 0.001), and T cells (*R* = 0.76, *p* < 0.001) (Appendix A).

### 3.6. Decreased CD8 Cells and Elevated M2 Macrophage Polarization in BCBM

To validate the gene expression profiles of immune cells in the TME, we performed IHC on FFPE tissues. CD86 was used as a marker of M1 macrophages, and CD163 and CD206 as markers of M2 macrophages. IHC was also performed for CCL19, which was most significant in the DEG analysis (Figure 4a). Expression of immune-cell markers was evaluated using the digital pathology application QuPath [30]. CD8+ T cells and CD86+ M1 macrophages showed significantly higher levels in primary BC than in BCBM (*p* = 0.006 and *p* < 0.001, respectively) (Figure 4b). In contrast, CD163+ M2 macrophages and CD206+ M2 macrophages showed significantly higher levels in BCBM than in primary BC (*p* < 0.001 and *p* < 0.001, respectively) (Figure 4b). Individual-patient BC-BM pair analysis showed that CD8+ T-cell and CD86+ M1 macrophage numbers were higher in primary BC than in BCBM (*p* = 0.007 and 0.337, respectively). CD163+ M2 macrophage and CD206+ M2 macrophage numbers were significantly higher in BCBM than in primary BC (*p* < 0.001 and *p* < 0.001, respectively) (Figure 4c). The BC-BM pair ratios of CD86+ M1 to CD163+ M2 macrophages and CD86+ M1 to CD206+ M2 macrophages were also significantly higher in primary BC than in BCBM (*p* = 0.005 and *p* = 0.008, respectively) (Figure 4d). Consistent with DEG analysis, CCL19 expression was significantly higher in primary BC than in BCBM (*p* < 0.001) (Figure 4e). When only HER2-positive subtype cases were selected, the unpaired Mann–Whitney U test showed that CD163+ M2 macrophage and CD206+ M2 macrophage were significantly higher in BM (*p* = 0.001 and *p* = 0.003, respectively) (Appendix A). The BC-BM pair ratios of CD86+ M1 to CD163+ M2 macrophages and CD86+ M1 to CD206+ M2 macrophages in the HER2-positive subtype tended to be higher in primary BC than in BCBM (*p* = 0.1 and *p* = 0.085, respectively) (Appendix A). CCL19 expression was significantly higher in primary BC than in BCBM with the HER2-positive subtype (*p* < 0.001) (Appendix A). When the analysis was restricted to the triple-negative subtypes, the CD8+ T cell count was significantly lower and the CD206+ M2 macrophage count was significantly higher in the BCBM (*p* = 0.03 and *p* = 0.03, respectively) (Appendix A). BC-BM pair ratios of CD86+ M1 to CD163+ M2 macrophages and CD86+ M1 to CD206+ M2 macrophages tended to be higher in primary BC than in BCBM with the triple-negative subtype (*p* = 0.13 and *p* = 0.082, respectively) (Appendix A). CCL19 expression also became significantly lower in BCBM than in primary BC with the triple-negative subtype (*p* = 0.042) (Appendix A). Pearson’s linear correlation analysis showed a trend toward an association between CCL19 expression and CD8+ T cells (*R* = 0.25, *p* = 0.097), and a significant association with CD86+ M1 macrophages (*R* = 0.31, *p* = 0.039) (Figure 4f). By contrast, CCL19 expression showed a tendency toward an inverse correlation with CD163+ M2 macrophages (*R* = −0.29, *p* = 0.06) and a significant inverse correlation with CD206+ M2 macrophages (*R* = −0.42, *p* = 0.0048).

Next, we assessed the prognostic significance of immune-cell profiles in terms of survival (Appendix A). A high CD8+ T-cell count was associated with longer recurrence-free survival (RFS), progression-free survival (PFS), and OS (*p* = 0.318, *p* = 0.029, and *p* = 0.153, respectively). A high CD86+ M1 macrophage count was also associated with a favorable RFS (log-rank *p* = 0.044) and PFS (log-rank *p* = 0.076), but not OS (log-rank *p* = 0.718). A high CD163+ M2 to CD86+ M1 macrophage ratio showed a tendency toward an association with worse RFS, PFS, and OS (*p* = 0.065, *p* = 0.155, and *p* = 0.059, respectively).

## 4. Discussion

We investigated the tumor, TME, and immune features of primary BC and BCBM. We found that when primary BC metastasizes to the brain, immune-related pathways are downregulated, and tumor intrinsic pathways are upregulated. Therefore, the immune environment of BCBM showed decreased tumor-suppressive components, such as cytotoxic CD8 T cells and M1 macrophages, and increased immunosuppressive M2 macrophages, compared to the immune environment of primary BC. The immune TME in BCBM is immunosuppressive compared to primary cancer. Ogiya et al. reported that BCBM had significantly fewer CD4+ cells, CD8+ cells, and FOXP3+ cells compared to primary BC based on IHC of 46 pair-matched samples [33]. Zhu et al. observed significantly lowered immune scores by ESTIMATE in BCBM compared to primary BC, by RNA sequencing of 50 pairs of patient-matched samples [34]. They also showed that most immune-cell populations were significantly smaller in metastatic tumors, while M2-like macrophage populations were larger in metastatic tumors based on the GSVA score of Davoli and Tamborero [35,36] and abundance estimated from deconvolution methods (using CIBERSORT and TIMER) [37,38,39]. In addition, they showed elevated macrophage (CD68) and decreased B cell (CD20) and T cell (CD8) numbers in BCBM by multispectral IHC. The abovementioned reports are consistent with our findings. However, Lu et al. reported that plasma cell infiltration was significantly greater in BCBM than primary BC, and the M2 macrophage score was lower in BCBM than primary BC, as obtained by CIBERSORT microarray analysis (GSE76714, GSE125989, and GSE43837) of the GEO database [40]. There was no significant difference in most immune-cell scores, including CD8+ T cells and M1 macrophages, between primary BC and BCBM based on only in silico data analysis. 

We validated our data by IHC of 76 sample tissues, including 24 paired samples. The infiltration of CD8+ T cells and M1 macrophages was decreased, and that of M2 macrophages was increased in BCBM compared to primary BC. Similar results were reported for cancers other than BCBM. Song et al. showed that most immune-cell populations were decreased in lung cancer BM [41]. In that study, infiltration of macrophages and CD56dim-NK-cells was increased in lung cancer BM, similar to the increased CD163-positive M2 to iNOS-positive M1 macrophage and NCR1-positive NK cell to CD3-positive T-cell ratios. Jeong et al. reported that an increased population of tumor-associated macrophages (TAMs), including M2 macrophages, was associated with progression of primary BC and an unfavorable DFS [42]. Our results showed that an increased M2 population in BCBM was associated with decreased RFS, PFS, and OS. 

In addition, we performed immune cell profile analysis for the HER2-positive subtype and the triple-negative subtype separately. In the HER2-positive subtype, M2 macrophages were significantly higher in BCBM (Appendix A). Antibody-dependent cell phagocytosis mediated by macrophages has been reported to be the main cause of the effectiveness of trastuzumab, a HER2-targeting antibody [43]. Trastuzumab resistance was also overcome by phenotypic transformation from M2 to M1 macrophages [44]. Therefore, a synergistic effect can be expected in BCBM if a combinatorial treatment of trastuzumab and M2 to M1 conversion is applied to the HER2-positive subtype. Current treatment options for TNBC patients include a combination of surgery, radiation therapy, and/or systemic chemotherapy [45,46,47]. FDA-approved therapies that target the DNA damage repair mechanism of TNBC, such as PARP inhibitors, have shown minimal clinical benefit yet [48,49]. A recent clinical precedent has been established by FDA approval for two TNBC immunotherapies, including an antibody-drug conjugate and an anti-PD-L1 agent [50,51]. The discovery of six molecular subtypes of TNBC, one of which is an immunomodulatory subtype, further accelerated the development of immunotherapeutic strategies for this disease indication [52]. In addition, chimeric antigen receptor (CAR)-T cell therapy, a type of adoptive cell therapy that combines the antigen specificity of an antibody with the effector function of T cells, has emerged as a promising immunotherapy strategy to improve the survival rate of TNBC patients [50,53]. In our study, in the triple-negative subtype, CD8+ T cells were significantly lower in BCBM than in primary BC. Oshi et al. showed that a high CD8 T-cell score was associated with a good prognosis in triple-negative primary breast cancer [54]. Enhancement of CD8+ T cell can be set as a therapeutic goal in the triple-negative subtype of BCBM. 

TAMs in the TME can polarize toward the M1 (proinflammatory) or M2 (anti-inflammatory) phenotype in response to local stimuli, such as cytokines [55,56]. M2-type TAMs promote cancer progression by a variety of mechanisms, including inflammation-induced tumor initiation, immune evasion, and immunosuppression, thereby promoting subsequent tumor growth and metastasis [56,57]. Several approaches have been developed to therapeutically target TAMs. They involve suppressing monocyte recruitment into the TME, blocking M2 polarization, and suppressing proinflammatory cytokines and other stimuli responsible for chronic inflammation in the TME [56,58]. Inhibitors of CSF1R, CCL2 and CCR2, CD47/SIRPα complex antagonists, CD40 agonist antibodies, and inhibitors of PI3Kγ and TREM2 protein are undergoing clinical evaluation for various tumor types [58]. Macrophage-targeted therapeutics may enhance antitumor efficacy by increasing cross-presentation to CD8+ T cells [56]. We found a difference in the expression of CD8+ T cells and M1 and M2 macrophages in the TME between primary BC and BCBM. Our findings will facilitate assessment of immunotherapies, especially those targeting M2 macrophages in BCBM.

CCL19 and CCL21 are C–C chemokine ligands that bind to the chemokine receptor CCR7. CCL19 and CCL21 are mainly secreted by reticular stromal cells in lymphoid organs, and are considered critical immune modulators [59]. Human B cells, expanded T cells, and dendritic cells express CCR7; its expression on peripheral T cells induces their migration by binding with its cognate ligands CCL19 and CCL21 [60,61,62]. The alteration of the T-cell population from dense infiltration in primary BC to sparse distribution in BCBM observed in this study may be mediated by altered expression of CCL19 and CCL21 during metastasis. However, because *CCR7* was not included among the 770 genes of NanoString nCounter PanCancer IO360^TM^, we could not fully assess the CCL19/CCL21-CCR7 axis. Chemokine ligands and their receptors play important roles in cancer biology, including immune-cell recruitment, tumor-cell proliferation and apoptosis, and metastasis [63]. Otero et al. reported that stimulation of CCR7 with both CCL19 and CCL21 induced G-protein activation, the ERK1/2 signaling pathway, calcium mobilization, and cell migration [64]. Xu et al. reported that decreased CCL19 induced BC-cell proliferation, migration, and invasion in vitro [65]. Liu et al. reported that expression of CCL21 by IHC was higher in lymph node metastatic cancer than primary BC [66] and Mamoor et al. showed that *CCL21* expression was significantly higher in the lymph nodes of patients with metastatic BC using microarray data, GSE10893 and GSE124648 [67,68,69]. Moreover, *CCL21* was not differentially expressed in four BCBMs compared to the primary tumor of the breast. In comparison, we found that *CCL19* and *CCL21* expression in 12 BC-BM pairs was significantly reduced in BCBM compared to primary BC. Furthermore, the CCL19 protein level in 24 BC-BM pairs was significantly lower in BCBM than primary BC. To our knowledge, there have been no reports on the functions of CCL19 and CCL21 in BCBM. The roles of CCL19 and CCL21 in the altered immune environment of BCBM warrant further investigation. 

BC is a heterogeneous disease classified into molecular subtypes based on ER, PR, and HER2 expression analyzed by IHC and in situ hybridization [70,71]. HR expression of metastatic lesions does not always reflect that in the primary tumor [72]. BCBM can have discordant hormone or HER2 expression compared to the corresponding primary tumor. Gaedcke et al. first reported the tendency to lose HR expression and to gain HER2 amplification in BCBM [73]. Palmieri et al. reported an experimental model in which HER2 overexpression promotes proliferation of metastatic tumor cells in the brain [74]. Lee et al. showed a trend toward increased HER2-enriched PAM50 subtypes in BCBM using a Nanostring nCounter analysis in 20 primary BC and 41 BCBM samples [75]. Schrijver et al. meta-analyzed a series of previous research on receptor mismatches in metastatic BC and showed that ER conversion occurred in 20.8% of the cases, PR conversion in 23.3%, and HER2 conversion in 12.5% during the metastasis to the central nervous system [76]. Priedigkeit et al. also reported the expression changes of clinically actionable genes in the majority of patients, emphasizing the gain of HER2 expression in around 20% of baseline HER2-negative tumors [77]. In the most recent systematic review, Alexander et al. reported 8% discordancy in ER, PR, and HER2 between primary BC and BCBM [78]. In addition, from primary BC to BCBM, there was a trend toward the triple-negative and HER2-positive subtype, and a trend away from the ER- and HER2-positive subtype. In this study, the discordancy of ER was 19.2% (5/26), while that of HER2 was 15.4% (4/26); the subtype change rate was 11.5% (3/26) based on IHC. Moreover, in all cases of HER2 discordancy, the HER2 protein level was increased in BCBM. We also showed that *ERBB2* gene expression was increased, and *ESR1* gene expression decreased, in BCBM compared to primary BC (by NanoString assay). Taken together, our findings were consistent with previous studies. Biopsies for brain metastases are not always performed in routine clinical practice because metastatic brain lesions are considered to be limitedly accessible. With an assumable situation that the primary breast cancer was a triple-negative subtype but the BCBM turns out to be a HER2-positive subtype, the effectiveness of the current treatment needs to be modified. For flexible application of treatment modalities, it is necessary to perform biopsy on brain metastatic lesions and to confirm the molecular and immunological characteristics. 

The main strengths of the present study are that, for pairwise comparison to the tumor microenvironment of primary breast cancer and brain metastases, we analyzed immunological factors, tumor intrinsic factors, and stromal factors on top of the differentially expressed genes. The present study is the first report to show that tumor intrinsic factors such as cell proliferation, epigenetic regulation, hypoxia, metabolic stress, DNA damage repair, apoptosis, and autophagy were up-regulated, and stromal factor such as matrix remodeling and metastases was down-regulated in BCBM compared to primary BC. Most of the genes related to tumor inhibitory mechanisms or available immune-modulating agents were decreased in BCBM, and there was a significant decrease in the cancer immune cell population. The increase in tumor proliferation and glycolic activity in BCBM reflects suppression of the immune response for tumor growth promotion. Other tumor-intrinsic factors—such as epigenetic regulation, hypoxia, metabolic stress, DNA damage repair, apoptosis, and autophagy—were upregulated in BCBM. We also confirmed the results of transcriptome analysis by immunohistochemistry on tissue samples from 50 patients with BCBM. However, the greatest limitation of the study is the small number of cases used for transcriptome analysis (*n* = 12 pairs), which places some objective limitations on the generalization of the results and prevents further sub-group analysis. 

## 5. Conclusions

In summary, brain metastatic lesions enable immune escape during BCBM, and tumor-intrinsic factors are also involved in tumor proliferation and immune suppression. These functional shifts are likely mediated by waning activity in the CCL19/CCL21/CCR7 axis. Although the number of matched samples was small and in vitro laboratory studies of functional mechanisms were not performed, potential therapeutic targets for BCBM were comprehensively assessed. An enhanced understanding of the altered immunosuppressive properties and tumor-intrinsic factors during BCBM will enable formulation of therapeutic strategies for patients with BCBM.

## Figures and Tables

**Figure 1 cancers-13-04895-f001:**
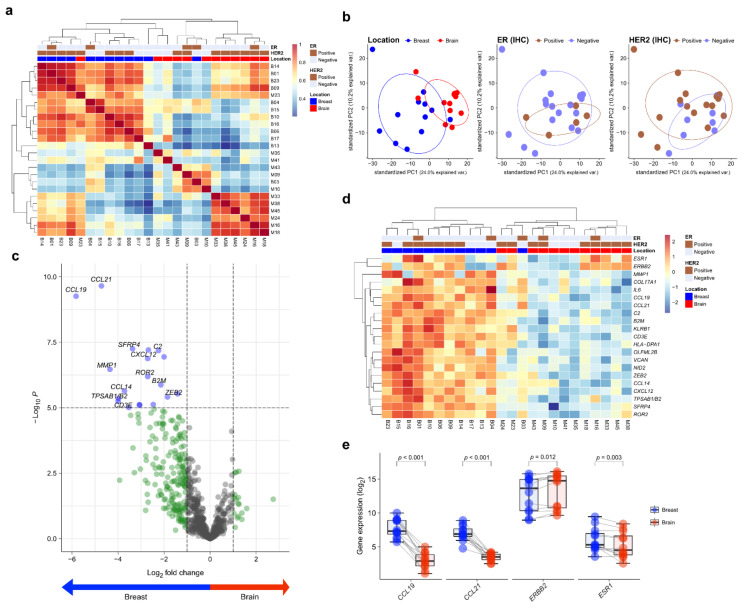
Differentially expressed genes (DEGs) in primary breast cancer and brain metastasis (BCBM) revealed by the nCounter^®^ IO360^TM^ Panel. (**a**) Correlation heatmap of unsupervised hierarchical clustering from 12 primary breast cancers and 12 matched brain metastases (BM) based on the gene expression profile. (**b**) Principal component analysis (PCA) plot of the 24 samples according to the involved location, estrogen receptor (ER) status, and human epidermal growth factor receptor 2 (HER2) status. (**c**) The volcano plot shows DEGs based on location with fold changes in expression >2 and *p* < 0.05. (**d**) Heatmap of representative genes showing differential expression profiles in primary BCBM. (**e**) Pairwise box plots of differential expression according to location showed that *ESR1*, *CCL19*, and *CCL21* expression was higher in primary breast cancer, while that of *ERBB2* was higher in BM.

**Figure 2 cancers-13-04895-f002:**
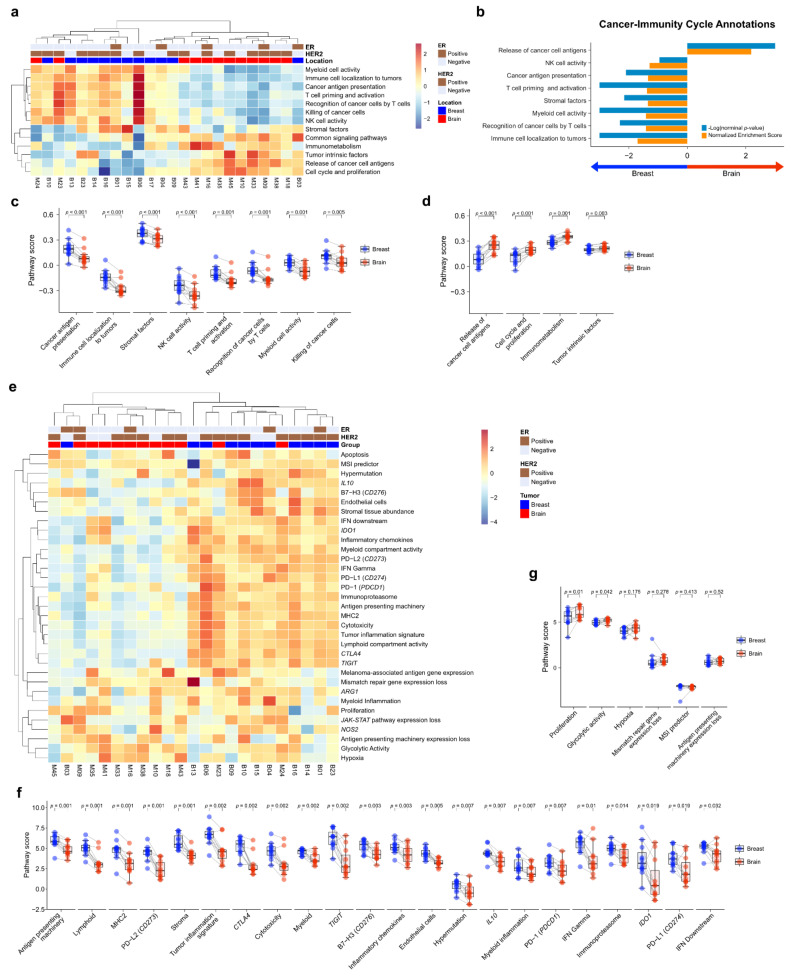
Pathway analysis of primary breast cancer and brain metastasis (BCBM). (**a**) Pathway scores for the cancer-immunity cycle annotation gene set from the nCounter^®^ PanCancer IO 360™ Panel were calculated with single-sample GSEA (ssGSEA) and visualized as a heatmap. Scores were Z-transformed and displayed on the same scale. (**b**) Gene set enrichment analysis of primary breast cancer versus brain metastasis (BM) visualized using the gene sets for cancer-immunity cycle annotation. (**c**,**d**) Pairwise box plots of cancer-immunity cycle annotation according to location, showing the sets of pathway scores that were downregulated in BM (**c**) and upregulated in BM (**d**). (**e**–**g**) PanCancer IO 360 biological signatures calculated using the nSolver analysis program. (**e**) Heatmap of PanCancer IO 360 Biological Signatures revealed that most immune-related scores were downregulated in BM. (**f**,**g**) Pairwise box plots of PanCancer IO 360 biological signatures according to location showing the sets of pathway scores downregulated in BM (**f**) and upregulated in BM (**g**).

**Figure 3 cancers-13-04895-f003:**
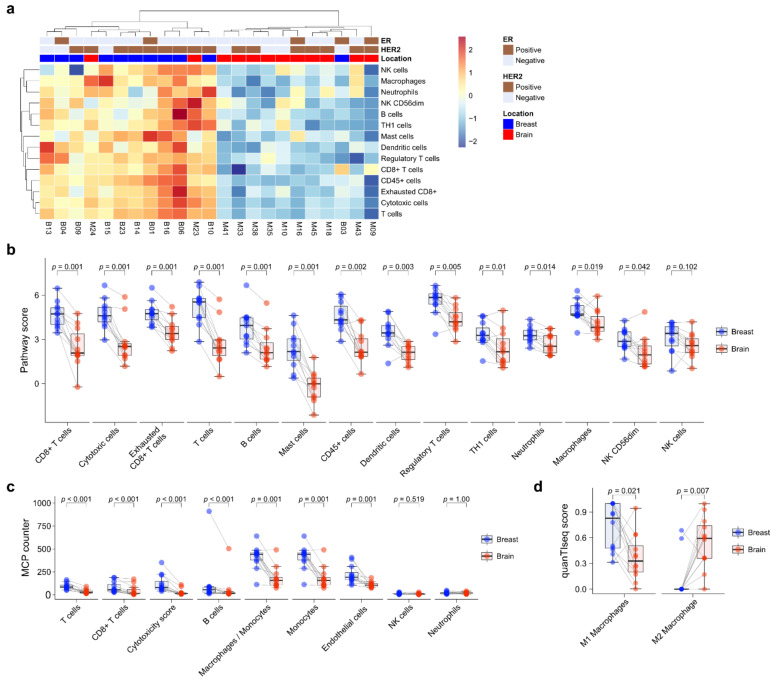
Immune-cell profile analysis of primary breast cancer and brain metastasis (BCBM) using the nCounter^®^ PanCancer IO 360™ Panel. (**a**) Heatmap of immune-cell profiling showing separate clusters in primary breast cancer versus BM. (**b**) Pairwise box plots of immune-cell scores obtained using the NanoString nSolver program according to tumor location. (**c**) Pairwise box plots of immune-cell scores obtained using the MCP counter method according to tumor location. (**d**) Pairwise box plots of M1 and M2 macrophage scores according to tumor location obtained using quanTIseq.

**Figure 4 cancers-13-04895-f004:**
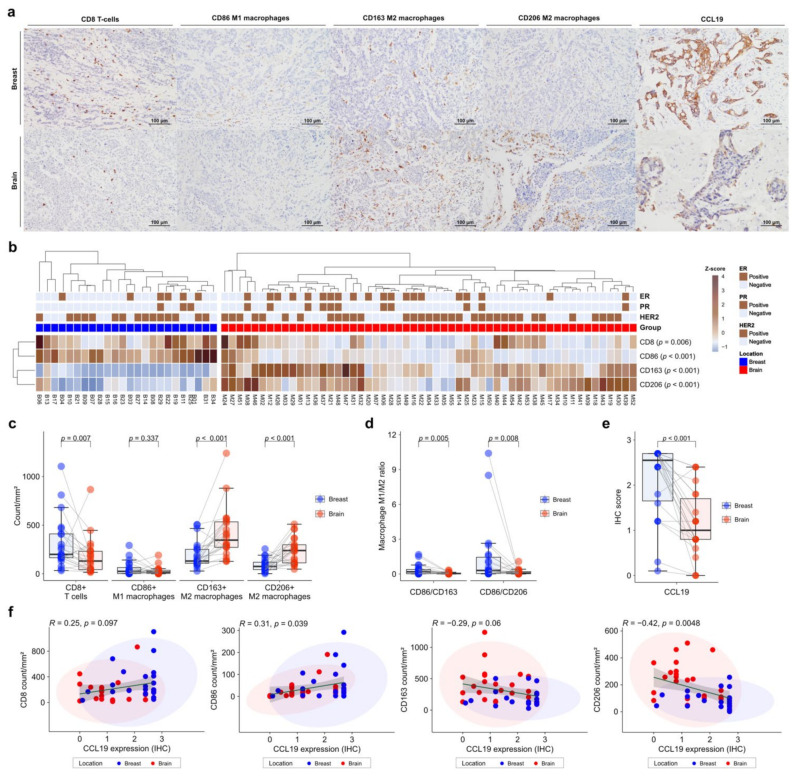
Immunohistochemistry (IHC) analysis of immune-cell profiles in primary breast cancer and brain metastases (BCBM). (**a**) Representative histograms of IHC on primary BCBM. From top to bottom, primary BCBM (original magnification, ×200). From left to right, immunohistochemical staining for CD8, CD86, CD163, CD206, and CCL19. (**b**) Heatmap of the IHC count of CD8+, CD86+, CD163+, and CD206+ cells in 55 cases of BM and 24 primary breast cancers according to location. (**c**) Pairwise box plots of counts of CD8+ cells, CD86+ cells, CD163+ cells, and CD206+ cells according to tumor location. (**d**) Pairwise box plots showing the relative proportions of CD86+ cells per CD163+ cells or CD206+ cells according to tumor location. (**e**) Pairwise box plot of CCL19 expression in 24 matched pairs of primary breast cancer and BM samples according to tumor location. (**f**) Linear correlation analyses of CCL19 expression in association with CD8+, CD86+, CD163+, and CD206+ counts, visualized as dot-correlation plots.

**Table 1 cancers-13-04895-t001:** Baseline characteristics of the study population.

Parameter	Metric	Primary Breast (*n* = 44)	Brain Metastases (*n* = 50)
Age at diagnosis (y)	Median (range)	47 (34–70)	54 (38–76)
Time from diagnosis of primary to brain metastasis (y)	Median (range)	3 (0–19)	
Size (cm)	Median (range)	2.4 (0.4–40)	4.15 (2.3–6.7)
ER status	*n* (%)		
	Positive	20 (45.5%)	17 (34.0%)
	Negative	24 (54.5%)	33 (66.0%)
PR status	*n* (%)		
	Positive	16 (36.4%)	9 (18.0%)
	Negative	28 (63.6%)	41 (82.0%)
HER2 amplification	*n* (%)		
	Positive	23 (52.3%)	25 (50.0%)
	Negative	21 (47.7%)	25 (50.0%)
Data available for NanoString	*n* (%)	12 (27.2%)	12 (24.0%)
Tissue available for IHC	*n* (%)	26 (59.0%)	50 (100%)

ER, estrogen receptor; PR, progesterone receptor; HER2, human epidermal growth factor receptor 2; IHC, immunohistochemistry.

## Data Availability

The datasets generated during and/or analyzed during the current study are available from the corresponding author on reasonable request.

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
