# Peer review of "Evolution of the Tumor Microenvironment toward Immune-Suppressive Seclusion during Brain Metastasis of Breast Cancer: Implications for Targeted Therapy"

_cancers, 2021, doi:10.3390/cancers13194895_

Round 1
Reviewer 1 Report
The study is well designed and well presented. My biggest concern is about the novelty of the work and the expansion of the discussion.
I recommend that the novelty of the work be explained in the paper.
The following papers are suggested to be read and used in the paper:
- PMID: 26527317
- doi: 10.1038/srep28623
- PMID: 27822408
- https://doi.org/10.1016/j.clbc.2016.08.008
Given the importance of TNBC and Her2 positive BM (especially TNBC due to lack of targeted therapy), please expand the discussion toward these subtypes.
Please expand the discussion towards subtype switching.
The importance of targeted therapy and treatment selection based on the results should be discussed in more detail.
Author Response
REVIEWER1
The study is well designed and well presented. My biggest concern is about the novelty of the work and the expansion of the discussion.
- I recommend that the novelty of the work be explained in the paper.
Answer) We appreciate the reviewer’s point. We have added some sentences on the strengths and limitations of the current study in the last paragraph of our discussion (line 524 – 548).
- The following papers are suggested to be read and used in the paper:
PMID: 26527317
doi: 10.1038/srep28623
PMID: 27822408
https://doi.org/10.1016/j.clbc.2016.08.008
Answer) Thank you for your suggestion on good references. We have taken a closer look at these two papers as suggested by the reviewer.
â‘ doi: 10.1038/srep28623
This paper appears similar to our study methodologically. Lee et al. performed differential gene expression analysis in 20 primary breast cancers and 41 brain metastases using Nanostring analysis. They used the PAM 50 gene set as a panel of 252 target genes. The PAM50 panel is aimed at subtype analysis of breast cancer. It differs significantly in scope from our study's NanoString nCounter PanCancer IO 360TM. The NanoString nCounter PanCancer IO 360TM is intended for the analysis of genes and signatures of known or potential clinical relevance with respect to tumors, microenvironment and immune responses, and contains 770 gene expression values. Therefore, the part we could use in Lee et al.'s study was the pseudotype change in brain metastasis. The tendency for HER2-rich PAM50 subtypes to increase in brain metastases was consistent with our study. We added this to “Discussion” (line 509-510). However, the results of differential gene expression analysis were far from our results, probably because there were not many overlapping genes between the gene panels used in the two studies.
â‘¡ https://doi.org/10.1016/j.clbc.2016.08.008
This paper also appears similar to our study methodologically. However, their project was a very different study from ours. Duchnowska et al. performed differential gene expression analysis using DASL microarrays in 119 advanced triple-negative primary breast cancer. And they retrospectively analyzed the occurrence of brain metastases according to the subtype of triple-negative breast cancer. They did not compare transcriptome analysis between brain metastases and primary breast cancer. In our study, 6 cases were triple-negative cancers with gene expression value information. For triple-negative cancer subtyping, we have accessed the triple-negative breast cancer subtyping site presented in this study (http://cbc.mc.vanderbilt.edu/tnbc/). However, we were not able to analyze subtyping. A sufficient number of genes must be matched for this subtyping analysis. The NanoString nCounter PanCancer IO 360TM panel used in our study contained only 770 genes. Therefore, subtype analysis of triple-negative cancer was not possible.
- Given the importance of TNBC and Her2 positive BM (especially TNBC due to lack of targeted therapy), please expand the discussion toward these subtypes.
Answer) We appreciate the reviewer’s comment. Since the extent of the discussion of TNBC and HER2 positive types covers a very wide area, it is necessary to focus and discuss the subtypes. Our study focused on the microenvironment and changes or differences in the immune response of BCBM, and we intended to highlight the aspects in our discussion on TNBC and HER2-positive types. We added Supplementary Fig. 4 to show differences in BCBM of immune cell profiles of HER2 positive subtypes and TNBC subtypes (line 364 – 378). And in the discussion section, the relationship between HER2-positive subtype target therapy (trastuzumab) and immune cells (macrophage) was further described (line 437 – 444). In addition, treatment methods for subtypes of TNBC and the prospects for immunotherapy of TNBC were additionally described (line 444 – 459). We believe that this supplementation has sufficiently expanded the discussion in the TNBC and HER2-positive subtypes.
- Please expand the discussion towards subtype switching.
Answer) Thank you for your comment. In the discussion section, we've expanded the treatment-associated aspect specifically for BCBM's subtype switching (line 507 – 516 and line 524 – 531). We believe that this supplement has sufficiently expanded the discussion of subtype conversion in our present study.
- The importance of targeted therapy and treatment selection based on the results should be discussed in more detail.
Answer) We appreciate the reviewer’s comment. Besides additional description on relationships between BC subtypes and immune profiles, the treatment options for subtypes of TNBC and the prospects for immunotherapy of TNBC were complemented (line 444 – 459).

Reviewer 2 Report
In their manuscript “Evolution of the Tumor Microenvironment Toward Immunosuppressive Seclusion During Brain Metastasis of Breast Cancer: Implications for Targeted Therapy " the Giun Noh et al. intended to explore the prospective treatment strategies for breast cancer brain metastasis (BCBM) by understanding the tumor microenvironment (TME) in BCMB.
However, there are a number of substantial issues that need to be resolved, listed below.
- Line no 79 & 80 is repetitive.
- The author should include the PR group comparison in Figures 1-3 and supplementary figures.
- It will be good if the author can include Telomere Integrity observation during BC advancement from primary to BCMB.
- Metastatic localization is not a random process; rather, it occurs at predetermined locations under the control of a plethora of microenvironmental, cellular, and molecular variables.
I think it would be preferable to provide the sequencing comparison of the driver mutation profile of the primary and metastatic samples for the betterment of the manuscript.
- However, the author stated that the study's limitation is the small number of matched samples, the authors should validate in-vitro observation in the future for a better understanding of the changed immunosuppressive features and tumor-intrinsic factors that occur during BMBC, which would facilitate pragmatically the development of therapeutic treatments for BCBM patients.
Author Response
REVIEWER2
In their manuscript “Evolution of the Tumor Microenvironment Toward Immunosuppressive Seclusion During Brain Metastasis of Breast Cancer: Implications for Targeted Therapy " the Giun Noh et al. intended to explore the prospective treatment strategies for breast cancer brain metastasis (BCBM) by understanding the tumor microenvironment (TME) in BCMB.
However, there are a number of substantial issues that need to be resolved, listed below.
- Line no 79 & 80 is repetitive.
Answer) Thank you for your scrupulous review. The first redundant sentence has been replaced with the following: “The TME is necessary for invasion, metastasis, and settling in a distant location” according to the citation (line 81-82).
- The author should include the PR group comparison in Figures 1-3 and supplementary figures.
Answer) We appreciate the reviewer’s comment. In the present study, all 12 pairs of cases subject to the NanoString nCounter PanCancer IO 360TM were negative for PR incidentally. We also considered expressing PR in all figures, but all negative PR results will give no information to the reader. In the heat map in Figure 4, the PR results can be seen in the IHC results of 76 cases. Compared to the frequency of ER positivity, PR was immunopositive in a smaller number of cases. We believe this can replace PR information that was not previously shown.
- It will be good if the author can include Telomere Integrity observation during BC advancement from primary to BCMB.
Answer) We appreciate the reviewer’s comment. In this study, we used 770 gene panels belonging to the NanoString nCounter PanCancer IO 360TM. This 770 genes panel contains MMR loss and MSI predictor information, but unfortunately does not include information related to telomere. We tried the gene set enrichment analysis (GSEA) using Msigdb. When all the gene sets related to telomere were selected, 32 out of 22596 gene sets in Msigdb 7.0v were related to telomere. The result is as follows.
(GO_CHROMATIN_SILENCING_AT_TELOMERE, GO_ESTABLISHMENT_OF_PROTEIN_LOCALIZATION_TO_TELOMERE, GO_MEIOTIC_ATTACHMENT_OF_TELOMERE_TO_NUCLEAR_ENVELOPE, GO_MEIOTIC_TELOMERE_CLUSTERING, GO_NEGATIVE_REGULATION_OF_TELOMERE_CAPPING, GO_NEGATIVE_REGULATION_OF_TELOMERE_MAINTENANCE, GO_NEGATIVE_REGULATION_OF_TELOMERE_MAINTENANCE_VIA_TELOMERASE, GO_NEGATIVE_REGULATION_OF_TELOMERE_MAINTENANCE_VIA_TELOMERE_LENGTHENING, GO_POSITIVE_REGULATION_OF_ESTABLISHMENT_OF_PROTEIN_LOCALIZATION_TO_TELOMERE, GO_POSITIVE_REGULATION_OF_TELOMERE_CAPPING, GO_POSITIVE_REGULATION_OF_TELOMERE_MAINTENANCE, GO_POSITIVE_REGULATION_OF_TELOMERE_MAINTENANCE_VIA_TELOMERE_LENGTHENING, GO_PROTECTION_FROM_NON_HOMOLOGOUS_END_JOINING_AT_TELOMERE, GO_REGULATION_OF_TELOMERE_CAPPING, GO_REGULATION_OF_TELOMERE_MAINTENANCE, GO_REGULATION_OF_TELOMERE_MAINTENANCE_VIA_TELOMERE_LENGTHENING, GO_TELOMERE_CAP_COMPLEX, GO_TELOMERE_CAPPING, GO_TELOMERE_LOCALIZATION, GO_TELOMERE_MAINTENANCE_IN_RESPONSE_TO_DNA_DAMAGE, GO_TELOMERE_MAINTENANCE_VIA_RECOMBINATION, GO_TELOMERE_MAINTENANCE_VIA_SEMI_CONSERVATIVE_REPLICATION, GO_TELOMERE_MAINTENANCE_VIA_TELOMERE_LENGTHENING, GO_TELOMERE_ORGANIZATION, GO_TELOMERE_TETHERING_AT_NUCLEAR_PERIPHERY, REACTOME_DNA_DAMAGE_TELOMERE_STRESS_INDUCED_SENESCENCE, REACTOME_EXTENSION_OF_TELOMERES, REACTOME_PROCESSIVE_SYNTHESIS_ON_THE_C_STRAND_OF_THE_TELOMERE, REACTOME_TELOMERE_C_STRAND_LAGGING_STRAND_SYNTHESIS, REACTOME_TELOMERE_EXTENSION_BY_TELOMERASE, REACTOME_TELOMERE_MAINTENANCE, WIEMANN_TELOMERE_SHORTENING_AND_CHRONIC_LIVER_DAMAGE_UP)
We attempted to analyze using a table of 770 gene expression values belonging to the NanoString nCounter PanCancer IO 360TM. The GSEA program showed the following error message. “After pruning, none of the gene sets passed size thresholds.” This is because the 770 gene panel does not contain enough telomere-associated genes. It is impossible to analyze telomere integrity with NanoString nCounter PanCancer IO 360TM 770 genes.
Metastatic localization is not a random process; rather, it occurs at predetermined locations under the control of a plethora of microenvironmental, cellular, and molecular variables.
- I think it would be preferable to provide the sequencing comparison of the driver mutation profile of the primary and metastatic samples for the betterment of the manuscript.
However, the author stated that the study's limitation is the small number of matched samples, the authors should validate in-vitro observation in the future for a better understanding of the changed immunosuppressive features and tumor-intrinsic factors that occur during BMBC, which would facilitate pragmatically the development of therapeutic treatments for BCBM patients.
Answer) We appreciate the reviewer’s comment. Mutation analysis is an important part in the oncology research. We also agree that it is really important to perform sequencing comparisons of driver mutation profiles in primary and metastatic samples to understand tumor biology. In this study, we used a panel of 770 genes from the NanoString nCounter PanCancer IO 360TM. This is for analysis on gene expression values. It does not contain mutation information. Sequencing containing mutation information is whole genome sequencing, whole exome sequencing, or a targeted panel sequencing, which is beyond the scope of this study. The scope of this study was the tumor microenvironment and immune cell profile, which is a different topic from the study of mutations.
